# Structure–Chiral Selectivity Relationships of Various Mandelic Acid Derivatives on Octakis 2,3-di-O-acetyl-6-O-tert-butyldimethylsilyl-gamma-cyclodextrin Containing Gas Chromatographic Stationary

**DOI:** 10.3390/ijms242015051

**Published:** 2023-10-10

**Authors:** Levente Repassy, Zoltan Juvancz, Rita Bodane-Kendrovics, Zoltan Kaleta, Csaba Hunyadi, Gergo Riszter

**Affiliations:** 1Rejtő Sándor Faculty of Light Industry and Environmental Engineering, Institute of Environmental Engineering and Natural Science, Óbuda University, Doberdó út 6, H-1034 Budapest, Hungary; leventerepassy@gmail.com (L.R.); bodane.rita@rkk.uni-obuda.hu (R.B.-K.); 2Department of Organic Chemistry, Semmelweis University, Hőgyes Endre Street 7, H-1092 Budapest, Hungary; riszter.gergo@phd.semmelweis.hu; 3Pro-Research Laboratory, Progressio Engineering Bureau Ltd., Muhar Street 54, H-1028 Budapest, Hungary; hunyadics@gmail.com; 4Higher Education and Industrial Cooperation Centre, Institute of Chemistry, University of Miskolc, Egyetem út 1, H-3515 Miskolc, Hungary; 5Artificial Transporters Research Group, Institute of Materials and Environmental Chemistry, Research Centre for Natural Sciences, Eötvös Loránd Research Network, Magyar Tudósok körútja 2, H-1117 Budapest, Hungary

**Keywords:** chiral separations, cyclodextrin, GC selector, mandelic acid-based test compounds

## Abstract

Frequently, a good chiral separation is the result of long trial and error processes. The three-point interaction mechanisms require the fair geometrical fitting and functional group compatibility of the interacting groups. Structure–chiral selectivity correlations are guidelines that can be established via trough systematic studies using model compounds. The enantiorecognition of the test compounds was studied on an octakis 2,3-Di-O-acetyl-6-O-tert-butyldimethylsilyl-gamma-cyclodextrin (TBDMSDAGCD) chiral selector. In our work, mandelic acid and its variously substituted compounds were used as model compounds to establish adaptable rules for other enantiomeric pairs. The mandelic acid and its modified compounds were altered at both their carboxyl and hydroxyl positions to test the key interaction forces of the chiral recognition processes. Ring- and alkyl-substituted mandelic acid derivatives were also used in our experiments. The chiral selectivity values of 20 test compounds were measured and extrapolated to 100 °C. The hydrogen donor abilities of test compounds improved their chiral selectivities. The inclusion phenomenon also played a role in chiral recognition processes in several cases. Enantiomer elution reversals were observed for different derivatives of hydroxyl groups, providing evidence for the multimodal character of the selector. The results of our research can serve as guidelines to achieve appropriate chiral separation for other enantiomeric pairs.

## 1. Introduction

Enantiomers, as optical isomers, are asymmetric structures, which cannot superimpose with their mirror images. The members of an enantiomeric pair can show rather different biological effects, in spite of their very similar structures [1]. The Contergan scandal was a terrific and tragic example of the different biological effects of enantiomers. Contergan (Thalodimide), a sedative pill, was sold as racemic mixtures. Its (*R*) isomer was harmless, but the (*S*) caused serious birth defects. The possible biological differences between the members of enantiomeric pairs forced the authorities to introduce directives for using optically pure medicines [2]. These guidelines are expected to only commercialize one member of the enantiomeric pairs. The requirement of optically pure medicines renders the enantioselective analyses necessary. A chiral pure medicine may contain less than 0.1% of the other member of the enantiomeric pairs. Selling enantiomerically pure agrochemicals is also becoming more and more common. The enantiomer-based quality control of pure products and chiral selective decomposition of enantiomeric pairs make necessary their enantiomer-based selective analyses. Chromatographic methods are most frequently used to determine the enantiomeric ratio of the compounds with GC [3] HPLC, SFC, and CE. In direct chiral selective chromatography, the members of an enantiomer pair create energetically different temporary interaction complexes with the selector, resulting in distinct retention times.

Generally, the chiral selectivity (α) exponentially increases with the lowering of the temperature (T) in the GC practice. If the lnα−1/T relation is linear, the given chiral recognition mechanism only belongs to one chiral recognition mechanism [4,5]. The curved relationships show that the chiral separations belong to more than one chiral simultaneous recognition mechanism.

It is a widely accepted fact that no universal chiral selector exists in chromatography [5]. The chiral recognition requests exact fits with the interacting groups of the selector and the analytes. The chiral recognition mechanism requires fair geometrical arrangements and chemical appropriateness among the interacting groups. Most of the chiral separations can be explained using the three-point interaction model [6,7]. According to the simplified three-point interaction model, one of the enantiomers interacts with three interacting groups (points) of the selector. The other enantiomer can only interact with two interacting groups (points) of the selector. The three interactions create more stable temporary association with the selector than the two interactions. This interaction difference is the basis of chiral selective chromatography. The more stable association complex of one isomer results in higher retention than that of the less stable isomer. The interaction types can be Van der Walls, dipole–dipole, dipole-induced dipole, π–π, H-bond, ionic, inclusion and repulsive type interactions. A Hydrogen acceptor group of analytes (e.g., oxo) can temporarily connect to a hydrogen donor group (e.g., hydroxy) of the selector. A bulky substituent of the selector (e.g., tercier butyl) can mantle the interaction sites of the selector from interactions with the selector’s interaction points.

The chiral recognition depends on the ratio of the interaction energies between the members of an enantiomer. Even an only 0.014 kcal/mol interaction difference (α = 1.02) is enough for the baseline separations of the enantiomers using very efficient chromatographic techniques (GC and CE) [8].

The most widely used chiral selectors are cyclodextrins in gas chromatography and capillary electrophoresis practices [3,5,8]. Cyclodextrins (CDs) are cyclic oligosaccharide molecules, which contain six, seven, or eight D(+)-glucopyranose units, assigned with the Greek letters α, β, and γ, respectively [9]. The glucose units join with α-1,4-glycosidic linkages. The CDs can be modified via substitution on their hydroxyl groups with various functional groups (i.e., acyl, alkyl, hydroxyalkyl, carboxyalkyl, sulfoalkyl, and phosphate) to improve their chemical, geometrical, and solubility characters. The CDs can encapsulate suitable substances in their apolar cavities, forming noncovalent host–guest inclusion complexes. The CDs have extremely broad chiral recognition spectra for the following reasons [8]:

CDs have numerous non-uniform chiral centers (35 in a β-CD). CDs have twisted, truncated cone shapes instead of a regular, symmetrical cone [8]. Chemically, the glucose units repeatedly uniformly connect to one another in the two-dimensional representation of CDs. However, they are sterically different in three dimensions caused by the twisted structures. The lengths and directions of the bonds around the various chiral centers are different in steric arrangement [8].

The hydroxyl groups of CDs can be substituted by different functional groups to enhance the interaction potentials of underivatized, native CDs [8]. Chiral substituent groups (e.g., hydroxypropyl and naphthyl-ethyl carbamoyl) add extra chiral centers to CDs, further broadening their recognition spectra [8]. The substituents offer not only various interaction sites, but also modify the cones’ shapes.

Most CD derivatives are randomly substituted products and mixtures of an enormous number of isomers [8,9]. They differ in terms of the numbers and the positions of their substituents, and almost every isomer differs from the others in terms of their chiral recognition features. Moreover, one CD molecule can be substituted with more than one substituent group, further broadening its chiral recognition spectra [9].

The derivatized CDs have moderately flexible structures. CDs can change their shape to interact with analytes using the “induced fit” mechanism, making possible three-point interactions [9].

The ionizable CDs can change their selectivity spectra depending on their neutral or ionized states [8]. They can show different chiral recognition properties toward analytes according to their ionized states. Moreover, their chiral recognition properties are influenced by the ionized states of the sample molecules.

The types of background buffer or mobile phases (normal, reverse, and polar–organic) determine the chiral selectivity features of CDs [8].

CDs can separate enantiomers not just with carbon atoms as chiral centers, but also with phosphorus or sulfur chiral centers. CDs can also separate chiral molecules with planar or axial chirality [5].

CDs have moderate rigidity; therefore, they generally produce lower chiral selectivity values than rigid selectors (e.g., amino acids, cellulose, etc.) [6]. Their relatively low chiral selectivity values are often not enough for baseline separations on the moderately efficiently packed columns. This is why the packed column techniques use CD-based chiral selectors less frequently than cellulose- and amylose-based chiral selectors, which can show high selectivity values. On the other hand, CDs are the most popular chiral selectors, using capillary columns with high efficiency values [8]. Several hundred thousand theoretical plate efficiency values of these techniques offer resolutions of enantiomer pairs with chiral selectivity values of only 1.01 [4]. Approximately 100 CD derivates have been successfully applied as chiral selectors in GC. Recently, the used CD-based stationary phases have been mixtures of moderately polar silicon polymer matrix and 15–50% persubstituted CD derivative chiral selectors in GC [4,8].

The flexible structures of cyclodextrins make it very hard to predict the chiral recognition while using these stationary phases. A successful chiral resolution is the result of a trial-and-error method on several occasions. The elution reversal of the optical antipodes can be observed using acetyl or trifluoro acetyl derivatives of analytes [10]. The CDs are multimodal selectors. They can interact in different ways with the analytes; even enantiomer reversals were recognized along a homologue series of enantiomers [11].

On the other hand, certain guidelines can be established for the chiral recognition features of a CD-based selector with systematic studies using model compounds.

The 6-O-tert-butyldimethylsilyl cyclodextrin derivatives are frequently used in chiral-selective GC, showing chiral selectivity toward to very broad range of enantiomers [12,13,14,15]. In our study, a mixture of octakis 2,3-Di-O-acetyl-6-O-tert-butyldimethylsilyl-gamma-cyclodextrin (TBDMSDAGCD) and silicon polymer chiral stationary phase was chosen as the chiral selector. This stationary phase the shows electron donor acceptor π-π and H-bond interaction characteristics. Its cavity is ready for the inclusion of guest molecules. This mechanism can play an important role in the chiral recognition processes. Chiral recognition can also happen without inclusion in this phase [14]. The 6-O-tert-butyldimethylsilyl bulky groups prevent the cavity from including guest molecules from the direction of “primary hydroxyl rim” [14]. Moreover, the tert-butyldimethylsilyl group has a flip-flop character. It also has a self-included arrangement [14].

Mandelic acid derivatives were chosen as the model compound to figure out the depth of the chiral recognition characters of (TBDMSDAGCD), which can apply to chiral separations of other compounds.

Mandelic acid (Figure 1) has acidic, alcoholic, and aromatic functional groups and a rather rigid structure [16].

The various interaction possibilities of mandelic acid offer good chiral recognition characteristics; therefore, the mandelic acids and their substituted compounds were frequently used as model compounds for chiral separations [17,18]. The different interaction types, different derivatives, and various substituents of mandelic acid help to better know the chiral recognition mechanisms of TBDMSDAGCD. Mandelic acid has low volatility and high polarity; therefore, it must be derivatized for GC analyses. The different derivatives allow us to figure out the roles of the various original and derivative groups in the chiral recognitions. The roles of the inclusions can be tested for chiral selectivity with differently substituted phenyl groups (ortho, meta and para). The chiral selectivities and retention parameters of 20 differently substituted and derivatized mandelic acid type compounds were tested. Such a big number of systematically measured data offers a chance to draw conclusions regarding the roles of the different interactions in the chiral recognition processes.

There are several studies (NMR, X-Ray, docking, and molecular orbital calculations) dealing with the chiral selectivity–structure relationships of CDs, but they can only be used to a limited extent in gas chromatographic conditions; therefore, our results are based on chromatographic data.

The purpose of this study is to provide guidelines to find the chiral separation parameters for other enantiomeric pairs more efficiently and quickly than via the trial-and-error method. Our conclusions (structure–chiral selectivity correlations) could be used to solve several other chiral separation problems.

## 2. Results

In order to achieve better collation, the measured and calculated results (chiral selectivity and Kovats retention index values (RI) are summarized at 100 °C. Every separated enantiomeric pair showed linearity in lnα−1/T relationship; therefore, we could calculate, via linear extrapolation, its chiral selectivity, retention time, and RI index values at 100 °C. The linearity relationships also proved that the given separation only belonged to one chiral recognition mechanism.

### 2.1. Results of Mandelic Acid Derivatives Where Mandelic Acid Was the Starting Material

Those derivatives (**1**–**6**) were tested first, where the mandelic acid was the starting material before our derivatization procedures, as presented in Figure 2.

It was necessary to derivatize at least one functional group of mandelic acid to be gas chromatographed. The mandelic acid could not be analyzed without derivatization, because it has high boiling point (321.8 °C), and it is very polar. We determined the effects of our derivatizations of carboxylic and hydroxyl functions of mandelic acid on their chromatographic properties. Our aim was to find the best analytical forms and explore the elution orders of the optical isomers. The minor first elution order is important in the case of trace enantiomeric impurity determination [19]. Secondly, an eluting trace enantiomeric impurity can be overlaid by the tailing of the major isomer. The results of this group (**1**–**6**) are summarized in Table 1.

The data of Table 1 contain not only their identification numbers (ID.), states of the functional groups, abbreviations of derivatives, calculated chiral selectivity values, elution orders of isomers, and Kovats retention index (RI) values, but also their calculated boiling points [20]. The boiling points of derivatives were also attached, which were useful to determine the intensity of the inclusion phenomena.

The data in Table 1 well present the achieved chromatographic data of mandelic acid derivatives at 100 °C, where mandelic acid was the starting material. The identification numbers, structure names, and abbreviations of the tested compounds are shown in Figure 2.

Every tested molecule was successfully chirally separated from this group (**1**–**6**). The best selectivity values were 1.27 for mandelic acid ethyl ester (**5**, Et,OH) and 1.24 for mandelic acid methyl ester (**2**, Me,OH) at 100 °C. It seems that the unchanged, free hydroxyl functions offer good chiral recognition properties. Probably, the lone electron pairs of the chiral selectors made enantiomer selective H-bond interactions with the hydroxyl groups of mandelic acid alkyl esters.

The methyl ether derivative of mandelic acid (**1**, Acid,MeO) showed an only moderate chiral selectivity value of 1.14. Of course, the hydrogen atom of the carboxyl group also had H-donor properties. Its moderate selectivity could be explained by two reasons. The position of acidic hydrogen has different steric arrangements than the hydrogen of hydroxyl groups. The latter group better fitted the structure of the chiral selector. The moderate chiral selectivity of mandelic acid methyl ether (**1**, Acid,MeO) could also cause strong achiral adsorptive interactions with the test material. The strong achiral adsorptive interactions significantly increased the retention times of the free acidic isomers. The interaction differences coming from their chiral selectivity played a reduced role in the alpha values because the achiral interactions had superior roles in the absolute values of retention times [21]. The H-bond donor properties of these three enantiomers (**1**, **2**, and **5**) played key role in their chiral recognition. The mandelic acid methyl ether methyl ester derivative (**3**, Me,MeO) only produced a 1.01 chiral selectivity value because this derivative had no H-donor ability.

In the cases of **1**, **2**, and **5** (compounds with underivatized carboxyl or hydroxyl groups), the S enantiomer was eluted first, followed by the R enantiomer. Probably, they have the same chiral recognition mechanism. On the other hand, the acetate derivatives (**4**, Me,AcO and **6**, Et,AcO) show that R eluted first, before the S enantiomer did so (Figure 3). The retention orders of different derivatives could be caused by their different chiral recognition mechanisms. This phenomenon might come from the fact that the **4**, Me,AcO and **6**, Et,AcO (acetate derivatives) have no H-donor ability, but they can show electron donor ability with their acetate groups. It is interesting to note that the methyl ester methyl ether derivatives produced *S* before the **R** elution orders.

The different chiral selective mechanisms of acetate (**4**, Me,AcO and **6**, Et,AcO) and free hydroxyl derivatives (**2**, Me,OH and **4**, EtOH) also suggested their rather different lnα–1/T relationships (Figure 4). Moreover, the acetate derivatives (**4**, Me,AcO and **6**, Et,AcO) produced higher RI values than the free hydroxy derivatives (**2**, Me,OH and **4**, EtOH), in spite of the free hydroxy derivatives having higher boiling points and polarities.

The inclusions phenomena could influence the chiral selectivity to a moderate extent in the cases of compounds **1**–**6**. However, the temperature dependencies of selectivity of ethyl (**5**, EtOH) and methyl esters (**2**, MeOH) are different (Figure 5). The methyl ester derivate of mandelic acid showed higher chiral selectivity than the ethyl ester derivate of mandelic acid above 140 °C, but at lower temperature range, the opposite outcome was noted.

This difference in temperature dependency probably came from their different inclusion behaviors. The inclusions were more intensive at low temperatures, and the ethyl ester had higher chiral selectivity inclusion ability than the methyl ester derivative [9].

### 2.2. Results of Ring-Substituted Mandelic Acid Derivatives

In this subsection, we discuss the results of the derivatives of aryl-methoxy- and aryl-chloro-substituted mandelic acid. We had ortho (2)-, meta (3)-, and para (4)-substituted isomers from both substituents, but only from the chloro-substituted isomers did we have pure optical standards. The results are summarized in Table 2. The structures of tested derivatives are presented on Figure 6. 

Every tested ring-substituted mandelic acid (**7**–**15**) was efficiently chirally separated (α: 1.07–1.39). The free hydroxyl derivatives (**7**–**10**, **12**, **14**) gave high selectivity values. Probably, the H-donor abilities of these compound improved their chiral selectivity values on every occasion. The acetate derivatized molecules (**11**, **13**, **15**) showed significantly lower chiral selectivity values (1.07–1.10) than isomers with free hydroxyl groups (**10**, **12**, **14**), which showed a 1.12–1.39 chiral selectivity range. The ortho (**7**, 2MeO/Me,OH and **10**, 2Cl/Me,OH)-substituted isomers showed a lower chiral selectivity range (1.12–1.16) in both methoxy and chloro substitutions than the 1.24 selectivity value of mandelic acid methyl ester (**2**, Me,OH). The situation was different with the meta-substituted derivatives. The meta methoxy-substituted (**8**, 3MeO/Me,OH) derivative only gained a 1.10 α value. However, the meta chloro-substituted (**12**, 3Cl/Me,OH) derivatives reached an excellent selectivity of 1.27, which is higher than the value of **2** (Me,OH). Probably, the voluminous methoxy substitution better fitted in the meta position than the less voluminous chloro substitution fit in the cavity of TBDMSDAGCD. The relative retention between the orhto and meta isomers is smaller in the case of chloro substitutions than methoxy substitutions, showing that meta chloro substitution had good inclusion properties.

Table 2 not only contains the results of ring-substituted mandelic acid derivatives, but also includes the data of mandelic acid methyl ester (**2**, Me,OH) and its acetate (**4**, Me, AcO) in the interest of easy comparison.

The para ring-substituted derivatives produced protruding chiral selectivity values of 1.37 and 1.39, respectively, for the methoxy (**9**, 4MeO/Me,OH) and chloro (**14**, 4Cl/meOHM,) substituents. The differences between the chiral selectivity values are well demonstrated among the ortho, meta, and para isomers of chloro-substituted mandelic acid methyl esters in Figure 7.

Probably the protruding selectivity values of para-substituted derivatives came from the different inclusion abilities of differently positioned ring substituents. According to Figure 7, the steepest temperature dependence of para derivative (**14**, 4Cl/MeOH) could come from its most intensive inclusion [9]. The ortho (**7**, 2MeO/Me,OH and **10**, 2Cl/Me,OH)- and meta (**8**, 3MeO/Me,OH and **12**, 3Cl/MeOH,)-positioned isomers could not deeply immerse in the cavity of the TBDMSDAGCD chiral selector. These substituents laterally protruded, which prevented them from influencing the deep immersions. On the other hand, the para (**11**, 4MeO/Me,OH and **14**, 4Cl/Me,OH) substituents are in the axis of TBDMSDAGCD, which let them induce deep immersions. The deep immersions of the para isomers support higher RI values than the corresponding ortho and meta isomers.

In these cases, the inclusions played significant key roles in the chiral recognition processes. These findings all suggest that the phenyl moiety of the substituted mandelic acids is largely involved in the interactions.

The acetylated (**11**, **13**, **15**) derivatives also produced smaller chiral selectivity values than their free hydroxyl derivatives (**10**, **12**, **14**) in the case of chloro ring-substituted mandelic acid methyl esters. The smaller chiral selectivity values of acetate derivatives were caused by their decreased H-donor abilities. However, the tendencies of the decreases in the chiral selectivity of acetates are not same for ortho, meta, and para derivatives. The selectivity decrease was very significant between the free hydroxyl derivative (14, 4Cl/MeOH,) and its acetate derivative (**15**, 4Cl/Me,AcO) of para-substituted chloro derivatives. On the other hand, the selectivity decrease was small between the free hydroxyl derivative (**10**, 2Cl/Me,OH) and acetate derivative (**11**, 2Cl/Me,AcO) of ortho chloro-substituted derivatives. It is interesting to note the chiral selectivity of mandelic acid methyl ester acetate (**4**, Me,AcO) was higher than those of the chloro-substituted analogs (**11**, **13**, **15**).

The retention reversals of enantiomers were also recognized among the free hydroxyl and acetate derivatives in all cases of chloro ring-substituted derivatives. Probably, two types of derivatives belonged to different chiral recognition mechanisms, similar to the cases of unsubstituted mandelic acid (Figure 8).

### 2.3. Results of Mandelic Acid Derivatives Where Mandelic Acid Alkyl Chains or the Ring Systems Were Extended

Experiments were performed with alkyl chain-modified mandelic acid derivatives, which are summarized in Table 3. Unfortunately, we did not have optically pure isomers of these compounds; therefore, we only tested the methyl ester derivatives. The measured derivatives are as follows (Figure 9): phenyllactic acid methyl ester (**16**, +/+CH_2_Me,OH) and atrolactic acid methyl ester, (**17**, tercMe/MeOHq,). The phenyl ring of mandelic acid was also substituted with dioxolane gaining benzo[1,3]dioxol-5-yl-hydroxy-acetic acid methyl ester (**18**, 3,4 diox/Me,OH). We expanded the following aromatic ring systems: hydroxy-naphthalen-1-yl-acetic acid methyl ester (**19**, 1Naph/Me,OH) and Hydroxy-naphthalen-2-yl-acetic acid methyl ester (**20**, 2Naph/Me,OH), which were also tested.

The results of mandelic acid derivatives where mandelic acid alkyl chains or the ring systems were extended are summarized in Table 3.

The alkyl chains of the mandelic acid were modified to check the chiral selectivity changes if the elements surrounding of the asymmetric center were substituted. The 3-phenyllactic acid (**16**, +CH_2_/Me,OH) derivative had a methylene unit between the chiral center and phenyl ring, and the (**17**, qMe/Me,OH) had a methyl substitution on the asymmetric carbon atom. The system of abbreviations of derivatives are the same as in Figure 2. The structures of the molecules are shown in Figure 9.

The effects of aromatic ring expansion were also tested based on the change in chiral selectivity values. The **18** (3,4diox/Me,OH) refers to the dioxolane substitution of the 3,4 positions of the phenyl ring. Two naphthyl derivatives were also tested: hydroxy-naphthalen-1-yl-acetic acid methyl ester (**19**, 1Naph/MeOH) and hydroxy-naphthalen-2-yl-acetic acid methyl ester (**20**, 2Naph MeOH). The table also contains the data of the mandelic acid methyl ester (2, MeOH) for the sake of easy comparison.

The methyl substitution of mandelic acid alkyl chain had significant effects on their chiral selectivity values. One methylene unit substitution between the phenyl groups and chiral center nullified the chiral selectivity of TBDMSDAGCD toward the molecule (**16**, +CH_2_/Me,OH). No chiral selectivity was measured for this derivative, not even at a temperature as low as 70 °C. Its increased RI value showed that this compound had strong inclusion, without chiral recognition. Probably, the repulsion effect of longer alkyl chain was responsible for the eliminated chiral selectivity. It is well known from the literature that the alfa position of the aromatic functions from asymmetric centers of analytes generally has better selectivity than their beta analogs [5,8,11]. On the other hand, one methyl substitution on the asymmetric carbon atom improved the chiral selectivity of the atrolactic acid methyl ester (**17**, qMe/Me,OH) a lot. Its 1.407 chiral selectivity value was the highest among the tested materials. Probably, the increased rigidity of the surrounding of the chiral center was the main reason for this selectivity improvement. The repulsion effect of this quaternary methyl substitution may also play a role in this selectivity improvement. Molecule **17** (qMe/Me,OH) showed a smaller RI value than **2** (Me,OH), in spite of **17** (qMe/Me,OH) having one more methylene unit than **2** (Me,OH). This decrease in the RI value was caused by the shielding effect of quaternal methyl group on the polar hydroxy group, decreasing its H-donor ability. Mandelic acid methyl ester (**2**, Me,OH) probably had more intensive inclusion than **17** (qMe/Me,OH). The bulky chiral center of **17** (qMe/Me,OH) must have prevented its deep immersion in the cavity of the chiral selector.

Extending the ring system of mandelic acid (**18**–**20**) also caused chiral selectivity improvements.

In the case of benzo[1,3]dioxol-5-yl-hydroxy-acetic acid methyl ester (**18**, 3,4diox/Me,OH), the inclusions were more intensive with the function groups of TBDMSDAGCD. The 150 RI increase in **18** (3,4diox/Me,OH) was much higher than expected from the boiling point data. This fact suggested more intensive inclusions of **18** (3,4diox/Me,OH) compared to **2** (Me,OH). The RI difference should have been around 50 if only the boiling point data had been taken into consideration.

Both naphthyl analogs (**19**, 2Naph/Me,OH and **20**, 1Naph/Me,OH) had high chiral selectivity values showing intensive inclusion properties, which played roles in their chiral recognition mechanisms. The cavity of gamma CDs could perform more intensive interactions with the naphthyl compounds than phenyl compounds because the size of naphthyl compounds was better suited to the cavity sizes of the gamma CDs (18). Hydroxy-naphthalen-2-yl-acetic acid methyl ester (**19**, 2Naph MeOH,) made more intensive inclusions and had higher chiral selectivity values than hydroxyh-naphthalen-1-yl-acetic acid methyl ester (**20**, 1Naph/MeOH,). The role of the inclusion phenomenon was supported by a more than 200 RI value difference between the two naphthyl compounds, in spite of their calculated boiling points being almost the same. The 1-naphthyl substitution might show some repulsion effects in the cavity of the selector. Probably, the inclusion phenomenon a played key role in the recognition processes of the extended ring system derivatives of mandelic acid.

## 3. Materials and Methods

### 3.1. Materials

The mandelic acid and variously substituted mandelic acids (2-chloromandelic acid, 3-chloromandelic acid, 4-chloromandelic acid, 4-methoxymandelic acid, and 3,4-methylenedioxymandelic acid) were purchased from Sigma-Aldrich. 2-Methoxymandelic acid, 3-Methoxymandelic acid, 2-(1-Naphthyl)-2-hydroxypropionic acid, 2-(2-Naphthyl)-2-hydroxypropionic acid, phenyllactic, 2-methoxy-2-phenylacetic acid, mandelic acid methyl ether, and an atrolactic acid sample were generously donated by Prof. Karol Kacprzak (Department of Chemistry, Adam Mickiewicz University, Poznan, Poland). Methanol, ethanol, ethyl acetate, chloroform, anhydrous, Na_2_SO_4_, NaHCO_3_, HCl, H_2_SO_4_, acetyl chloride, and trifluoroacetic anhydride were products of Sigma-Aldrich. N-alkanes were products of Reanal Ltd. All of the applied chemicals were purissimum grade.

### 3.2. Instrumentation

The gas chromatographic experiments were performed via a Shimadzu 17A/QP5000 on-line coupled gas chromatographic/mass spectrometric instrument. A MEGA-DEX-DAC-Gamma (Mega S.r.l.) column (25 m × 0.25 mm with, d_F_ 0.25 μm) was used for separations. This column contained octakis 2,3-Di-O-acetyl-6-O-tert-butyldimethylsilyl-γ-cyclodextrin (TBDMSDAGCD) chiral selector in a moderately polar silicone polymer matrix. Helium was applied as a carrier gas.

A temperature controlled drying cabinet was used for the heat treatments during the derivatization processes. Reacti-Vials were purchased from Sigma-Aldrich.

### 3.3. Experimental Procedures

The low volatility and high polarity of mandelic acid made it necessary to derivatize one of its functional groups (or both) for its gas chromatographic analyses.

Preparing alkyl ester derivatives of acidic groups was carried out according to standard procedures [22].

In the cases when both isomers and racemic mixtures were available, 0.066 mmol racemic compounds and 0.033 mmol (*R*) isomer were mixed together in a Reacti-Vial and dissolved in 1 mL of methanol or ethanol. In this way, the determination of the retention order of *R* and *S* isomers was possible. One drop of concentrated HCl or H_2_SO_4_ was added as a catalyst. When we had only racemic mixtures, 0.1 mmol samples were taken. The closed vials were heated at 80 °C for 2 h. After the reaction mixtures were cooled down, they were quenched with 0.5 mL of distilled H_2_O, and 2 mL of ethyl acetate was added. The aqueous phases were neutralized with a 0.1 mol/L NaHCO_3_ solution. The phases were separated using syringes. The organic phases were dried with anhydrous Na_2_SO_4_, and the drained solutions were dried under N_2_ streams. The dried materials were dissolved in 3 mL ethyl acetate, and 1 mL of them were analyzed in GC/MS instrument.

Preparing acetate derivatives of alcoholic groups was carried out according to standard procedures [22].

Several esters were further derivatized to transform the hydroxyl groups into acetate. Next, 1 mL of each of the ethyl acetate solutions of the esters (0.033 mmol) was reacted with 90 μL (0.066 mmol) of acetyl chloride, which were added drop by drop into Reacti-Vials. The Reacti-Vials were closed and heated for 2 h at 80 °C. The reaction mixtures were cooled down and quenched with 0.5 mL of distilled H_2_O, and 2 mL of ethyl acetate was added. The aqueous phases were neutralized with a 0.1 mol/l NaHCO_3_ solution. The phases were separated using syringes. The organic phases were dried with anhydrous Na_2_SO_4_, and the drained solution was dried under N_2_ streams. The dried materials were dissolved in 1 mL ethyl acetate and analyzed via a GC/MS instrument.

We tested the chiral selectivity and Kovats retention indices [23] of different derivatives of mandelic acid. Alkyl chain-modified, ring-substituted, and ring-expanded mandelic acid compounds were also derivatized and measured as mandelic acid.

### 3.4. Measurements

We tried to make appropriate chiral separations, at least three, at different analysis temperatures. The three measured points were enough to calculate lnα−1/T curves, where α is the measured chiral selectivity and T is the absolute temperature (K) of the given separations. Our purpose was to gain α > 1.02 values in 5–120 min time intervals; however, it was not possible to achieve our goals on every occasion. In these cases, decreased analysis temperatures (longer retention times) were applied; otherwise, we could only achieve selectivity values lower than 1.02.

### 3.5. Calculations

For the sake of comparison, the measured and calculated chiral selectivity and Kovats retention index (RI) [23] values of different derivatives are given at 100 °C. The Kovats retention indices show the differences in the interaction energies of various derivatives better than their retention times. The chiral selectivity values and retention times were measured for every tested molecule at least three different temperatures. These data made it possible to count the ln α−1/T functional relationships and calculate the chiral selectivity values of tested compounds at 100 °C. The measured retention times (t_R_) of analytes also made it possible to count their RI values. If the enantiomers were separated, their mean retention times were the bases of RI calculations. The ln t_R_−1/T functional relationships were appropriate to calculate the retention times of analytes at 100 °C. The calculated and measured retention times of the tested derivatives were compared to the measured retention times of n-alkanes at 100 °C. The results of these comparisons gave the RI values of tested materials. We also calculated the boiling points [20] of different derivatives to gain information for the inclusion of the test molecules.

## 4. Conclusions

The derivatives of mandelic acid proved to be good materials to test the chiral selectivity of octakis 2,3-di-O-acetyl-6-O-tert-butyldimethylsilyl-gamma-cyclodextrin. Every isomer was successfully chirally separated, except for the 3-phenyllactic acid methyl ester from the 20 tested molecules. Generally, the H-bond donor ability of compounds (free carboxyl or hydroxyl functions) improved the chiral selectivity of the compounds. The H-bond acceptor properties of TBDMSDAGCD were one of the key interaction types of this selector. The chiral separations were, however, possible without H-donor functional groups, (e.g., methyl ether or acetate derivatives), but to a decreased extent. The chiral selectivity of ring-substituted derivatives highly depends on the position of their substituent. The para position showed the highest selectivity values, caused by their intensive inclusion. Probably, the inclusions played important roles in the chiral recognition processes on several occasions. Enantiomer reversal order was also observed among the derivatives, with free hydroxyl groups and compounds having acetate derivatives on the hydroxy functions. These reversal elution orders showed the multimodal characters of this chiral selector. On the other hand, a certain material only belonged to one chiral recognition process, having a linear lnα−1/T relationship.

Finally, the TBDMSDAGCD is recommended for chiral separation for the broad spectra of enantiomers because it has chiral selectivity features for different functional groups.

## Figures and Tables

**Figure 1 ijms-24-15051-f001:**
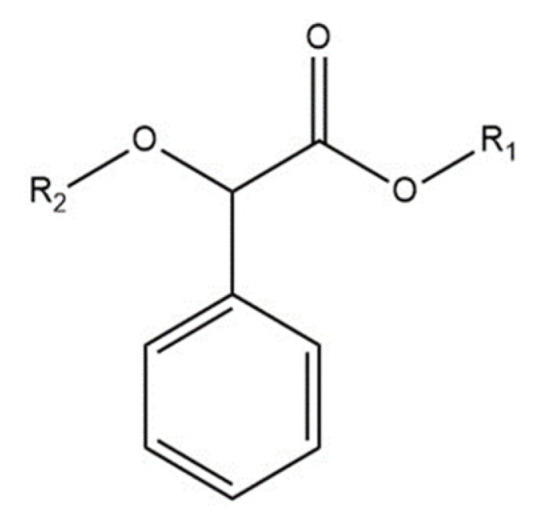
The structure of mandelic acid, where R_1_ and R_2_ are hydrogens. The R_1_ and R_2_ could be different in the tested molecules.

**Figure 2 ijms-24-15051-f002:**
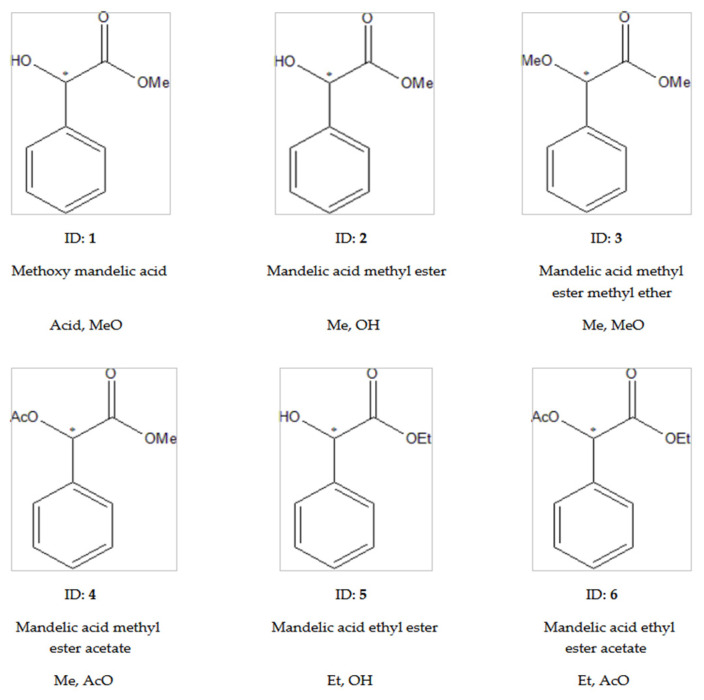
The tested derivatives of mandelic acid. This figure contains the structural formulas, identification numbers (bold), names derived from mandelic acid, and abbreviations of the tested compounds. The system of abbreviations are as follows: the first symbols refer to the states of carboxyl groups, while the second refers to the states of hydroxyl groups in the abbreviations. The symbols of acid groups are as follows: Acid, underivatized acid; Me, methyl esters; Et, ethyl ester. The abbreviation of hydroxyl groups are as follows: OH, underivatized hydroxyl; AcO, acetate; MeO, methyl ether. The * marks the asymmetric carbon atoms.

**Figure 3 ijms-24-15051-f003:**
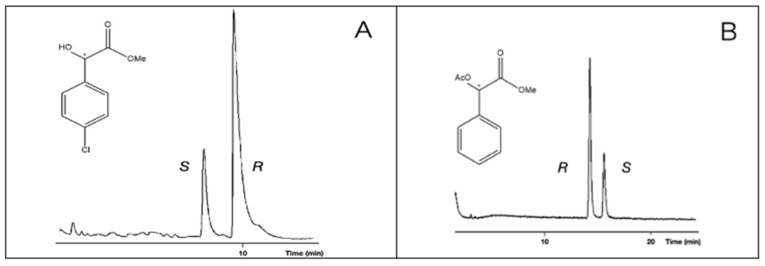
Reversal of elution orders between the enantiomers of mandelic acid methyl ester (**2**, Me,OH) in A chromatogram and mandelic acid methyl ester acetate (**4**, Me,AcO) enantiomers in B chromatogram. The samples had R isomer excesses. Conditions: instrument, Shimadzu 17A/QP5000 GC/MS; column, 25 m × 0.25 mm; stationary phase, MEGA-DEX-DAC-Gamma; carrier, Helium (50 cm/s); temperatures, (**A**) 130 °C and (**B**) 140 °C. The * marks the asymmetric carbon atoms.

**Figure 4 ijms-24-15051-f004:**
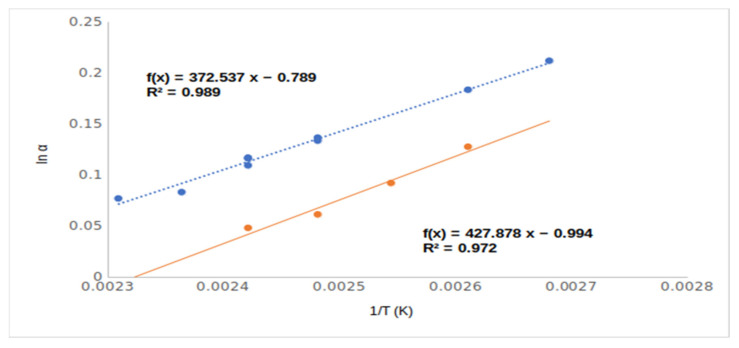
The chiral selectivity of mandelic acid ethyl ester (**5**, Et,OH) and mandelic acid ethyl ester acetate (**6**, Et,AcO) derivatives in the function of temperature (1/T). Symbols: (●) mandelic acid ethyl ester; (●) mandelic acid ethyl ester acetate. Gas chromatographic conditions were similar to those in Figure 3.

**Figure 5 ijms-24-15051-f005:**
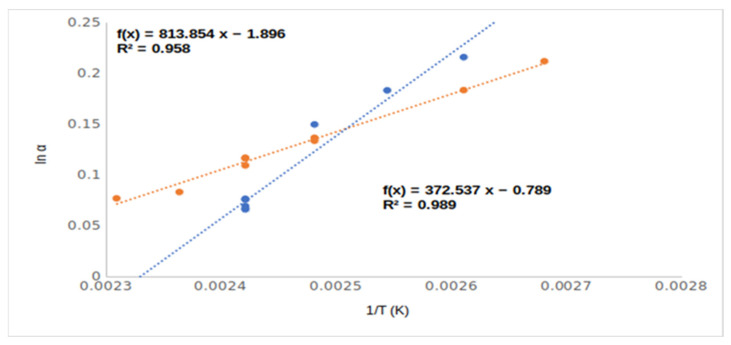
The temperature dependence of chiral selectivity values of mandelic acid methy ester (**2**, Me,OH) and mandelic acid ethyl ester (**5**, Et,OH). Symbols: (●) mandelic acid methyl ester (**2**, MeOH); (●) mandelic acid ethyl ester (**5**, EtOH). Gas chromatographic conditions were similar to those in Figure 3.

**Figure 6 ijms-24-15051-f006:**
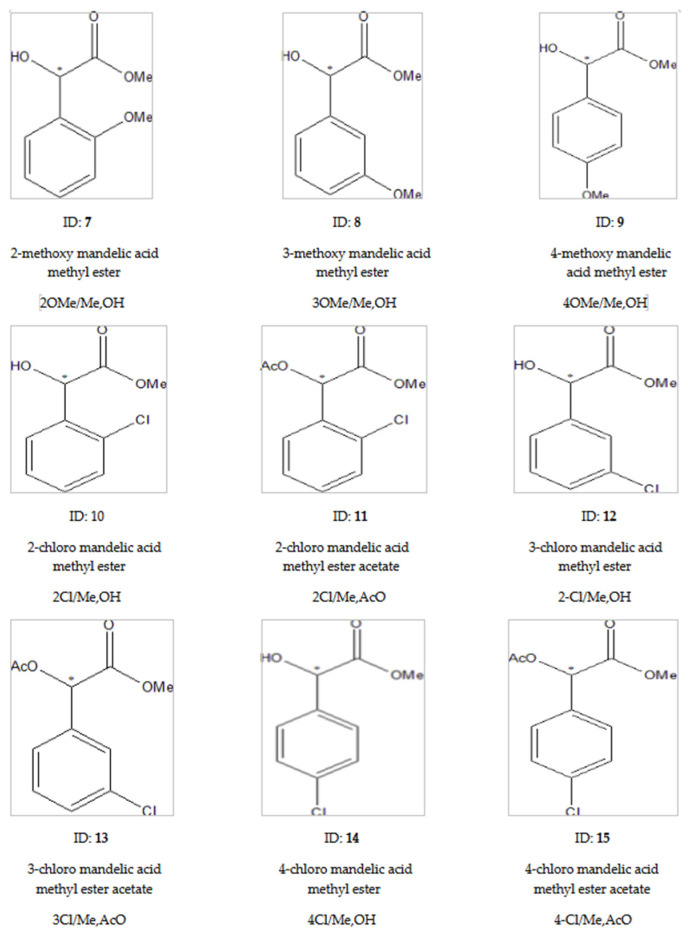
The tested ring-substituted mandelic acid derivatives. The ring substitutions are marked before the/symbols in the abbreviations. The ring substitution positions are assigned numbers: 2, ortho; 3, meta; 4, para, which are followed by the chemical types of substitutions: MeO, methoxy; Cl, chloro. The abbreviations of derivatives are the same as those presented in Figure 2 after the / sign. The * marks the asymmetric carbon atoms.

**Figure 7 ijms-24-15051-f007:**
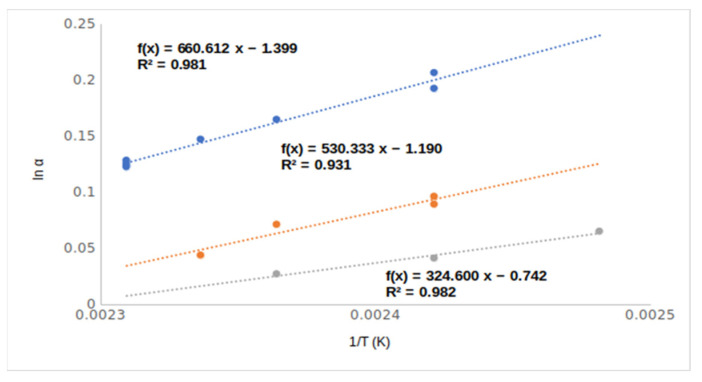
The temperature dependence of chiral selectivity values of chloro ring-substituted mandelic acid methyl esters. Symbols: (●) 2-chloro mandelic acid methyl ester (**10**, 2Cl/Me,OH); (●) 3-chloro mandelic acid methyl ester (**12**, 3Cl/Me,OH); (●) 4-chloro mandelic acid methyl ester (**14**, 4Cl/lMe,OH). Gas chromatographic conditions are similar to those in Figure 3.

**Figure 8 ijms-24-15051-f008:**
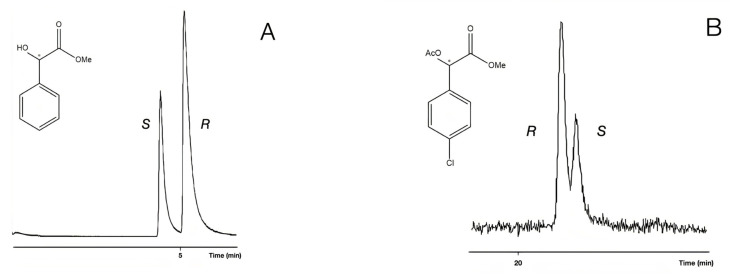
Reversal of the elution order between the enantiomers of 4-chloro mandelic acid methylester (**14**, 4Cl/Me,OH) in chromatogram A and 4-chloro mandelic acid methyl ester acetate (**15**, 4Cl/Me,AcO) enantiomers in chromatogram B. The samples had ***R*** isomer excesses. The gas chromatographic conditions were similar to the parameters of Figure 3, except for the temperatures: (**A**) 160 °C; (**B**) 130 °C. The * marks the asymmetric carbon atoms.

**Figure 9 ijms-24-15051-f009:**
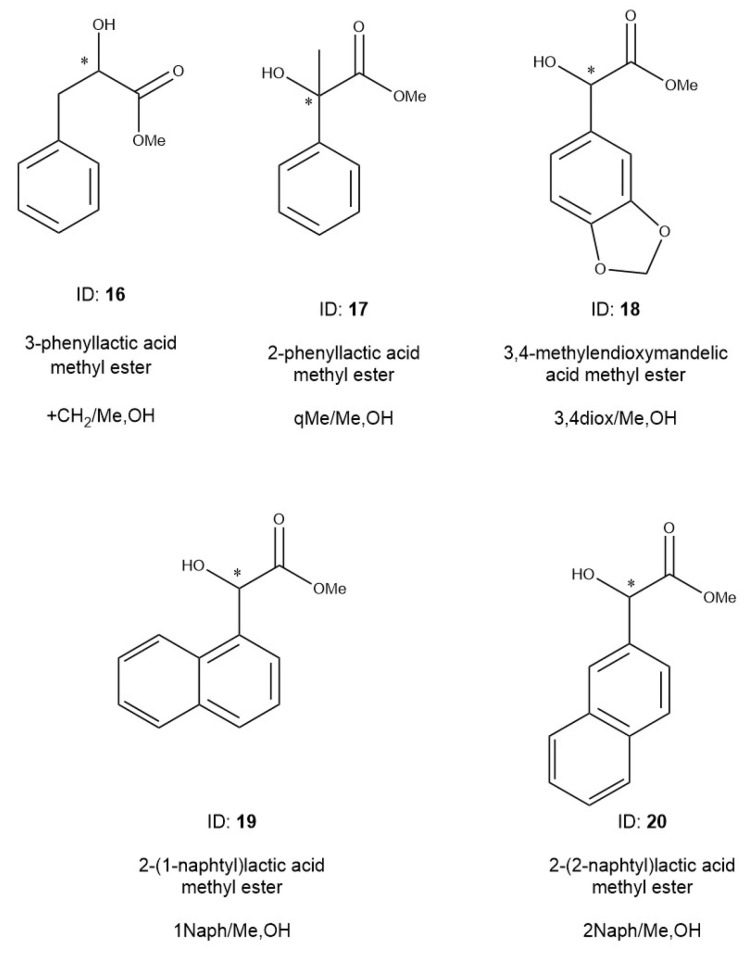
The structures, identification numbers, name, and abbreviations of alkyl- and ring-expanded mandelic acid methyl ester derivatives. The * marks the asymmetric carbon atoms.

**Table 1 ijms-24-15051-t001:** Results of mandelic acid derivatives where mandelic acid was the starting material.

ID. Number	Carboxyl R_1_ ^1^	Hydroxyl R_2_ ^1^	Abbreviation	Selectivity ^2^α (100 °C)	Elution Order ^3^	RI (100 °C)	Boiling Point [20]
**1.**	H	CH_3_	Acid,MeO	1.14	S	1901	289.51
**2.**	CH_3_	H	Me,OH	1.24	S	1770	266.08
**3.**	CH_3_	CH_3_	Me,MeO	1.01	S	1639	242.20
**4.**	CH_3_	OCHCH_3_	Me,AcO	1.17	R	1730	255.74
**5.**	CH_3_CH_2_	H	Et,OH	1.27	S	1735	282.27
**6.**	CH_3_CH_2_	OCHCH_3_	Et,AcO	1.06	R	1769	272.48

^1^ See Figure 1; ^2^ Chiral selectivity; ^3^ Firstly eluting isomer.

**Table 2 ijms-24-15051-t002:** Results of ring-substituted mandelic acid derivatives (**7**–**15**). The abbreviations and symbols are shown in Figure 6. The system of Table 2 is same as that in Table 1.

ID. Number	Carboxyl R_1_ ^1^	Hydroxyl R_2_ ^1^	Code	Selectivity ^2^ α (100 °C)	Elution Order ^3^	RI (100 °C)	Boiling Point [20]
**2.**	CH_3_	H	Me,OH	1.24	S	1770	266.08
**4.**	CH_3_	OCHCH_3_	Me,AcO	1.17	R	1730	255.74
**7.**	CH_3_	H	2OMe/Me,OH	1.16		1918	298.12
**8.**	CH_3_	H	3OMe/Me,OH	1.10		1890	298.12
**9.**	CH_3_	H	4OMe/Me,OH	1.37		1986	298.12
**10.**	CH_3_	H	2Cl/Me,OH	1.12	S	1887	291.72
**11.**	CH_3_	OCHCH_3_	2Cl/Me,AcO	1.10	R	1915	282.27
**12.**	CH_3_	H	3Cl/Me,OH	1.27	S	1884	291.72
**13.**	CH_3_	OCHCH_3_	3Cl/Me,AcO	1.08	R	1904	282.27
**14.**	CH_3_	H	4Cl/Me,OH	1.39	S	1905	291.72
**15.**	CH_3_	OCHCH_3_	4Cl/Me,AcO	1.07	R	1911	282.27

^1^ See Figure 1; ^2^ chiral selectivity; ^3^ firstly eluting isomer.

**Table 3 ijms-24-15051-t003:** The results of mandelic acid derivatives where the mandelic acid alkyl chains or the ring systems were extended. The structures, identification numbers, name, and abbreviations of alkyl- and ring-expanded mandelic acid methyl ester derivatives are presented in Figure 9.

ID. Number	Carboxyl R_1_ *	Hydroxyl R_2_ *	Abbreviation	Selectivity ** α (100 °C)	RI (100 °C)	Boiling Point [20]
**2.**	CH_3_	H	Me,OH	1.24	1770	266.08
**16.**	CH_3_	H	+CH_2_/Me,OH	<1.01	1829	282.27
**17.**	CH_3_	H	qMe/Me,OH	1.41	1727	268.34
**18.**	CH_3_	H	3,4diox/Me,OH	1.29	2048	325.41
**19.**	CH_3_	H	1Naph/Me,OH	1.18	1978	351.09
**20.**	CH_3_	H	2Naph/Me,OH	1.29	2209	351.09

* See Figure 1; ** chiral selectivity.

## Data Availability

The data presented in this study are available on request from the corresponding author.

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
