# Peer review of "Structure–Chiral Selectivity Relationships of Various Mandelic Acid Derivatives on Octakis 2,3-di-O-acetyl-6-O-tert-butyldimethylsilyl-gamma-cyclodextrin Containing Gas Chromatographic Stationary"

_ijms, 2023, doi:10.3390/ijms242015051_

Round 1

Reviewer 1 Report

The current manuscripts describe the study of the chiral recognition mechanism of mandelic acid derivatives and structurally some how similar compounds on a known chiral cyclodextrin phase, namely octakis 2,3-Di-O-acetyl-6-O-tert-butyldimethylsilyl-gamma-cyclodextrin (TBDMS- 24 DAGCD) chiral selector immobilised/bonded on silica (developed by others).

Line 39: This statement is not always correct “Enantiomers, optical isomers, are asymmetric molecules.”

Line 47: “Renders” rather than “makes”

Line 48: “Enantioselective” rather than “chiral selective”

Line 92: “Remove the dot”

Line 134 : The elution reversal of members of enantiomer” rephrase

Several analytes derivatisation did occur, rendering this process not viable regarding practicality. It is simply tedious, and enantiomeric purity might be affected through racemisation during derivatisation.

Not sure about the utility of gas chromatography in this process when HPLC can easily separate mandelic acid without any tedious workup or derivatisation. If it is to understand the mechanism of docking mode on derivatised gamma CD’s, this has already been known in lit. However, there might be some very limited novelty not supported by enough evidence though..

The author must show that the assumption/conclusion is correct by showing separations of other categories rather than structural similarity with the studied mandelic acid.

Moderate

Author Response

Thank you very much for the notes of reviwer.

My responses are in italics.

The current manuscripts describe the study of the chiral recognition mechanism of mandelic acid derivatives and structurally somehow similar compounds on a known chiral cyclodextrin phase, namely octakis 2,3-Di-O-acetyl-6-O-tert-butyldimethylsilyl-gamma-cyclodextrin (TBDMS- 24 DAGCD) chiral selector immobilised/bonded on silica (developed by others).

Line 39: This statement is not always correct “Enantiomers, optical isomers, are asymmetric structures” 

Response:  The text has been corrected: Enantiomers, optical isomers, have asymmetric structures 

Line 47: “Renders” rather than “makes”

Response: corrected

Line 48: “Enantioselective” rather than “chiral selective”

Response: corrected

Line 92: “Remove the dot”

Response: corrected

Line 134 : The elution reversal of members of enantiomer” rephrase

Response: The elution reversal of the peaks of enantiomer pairs can be observed..

Several analytes derivatisation did occur, rendering this process not viable regarding practicality. It is simply tedious, and enantiomeric purity might be affected through racemisation during derivatisation.

Response: The reviewer is right. Some reaction can cause racemization. However, our derivatization reactions are broadly used in the chiral GC, because they do not cause racemization. The high-resolution power of the GC can offer better separation method than HPLC in several occasions.

Not sure about the utility of gas chromatography in this process when HPLC can easily separate mandelic acid without any tedious workup or derivatisation. If it is to understand the mechanism of docking mode on derivatised gamma CD’s, this has already been known in lit. However, there might be some very limited novelty not supported by enough evidence though..

Response: Of course, mandelic acid can separated with HPLC and CE without derivatization. Our aim was to figure out the chiral separation characteristics of the octakis 2,3-Di-O-acetyl-6-O-tert-butyldimethylsilyl-gamma-cyclodextrin (TBDMS- 24 DAGCD) chiral selector. Our aim was not to make a successful separation method for mandelic acid. Mandelic acid and its derivatives was used as model compounds. Changing the functional groups of the tested compounds structure – chiral selectivity relationship was established. The flexible structure of octakis 2,3-Di-O-acetyl-6-O-tert-butyldimethylsilyl-gamma-cyclodextrin makes hardly predictable the from the docking studies. These docking studies are good to explain a successful chiral separation, than to estimate a good chiral separation of an enantiomeric pairs in advance.

A systematic chiral selectivity – structure relationship study was missing for TBDMS- 24 DAGCD chiral selector in the literature. Our aim was to fill this gap. Generalizing our results, the H-bond donor feature of the tested enantiomers advantageous in this selector. The chiral selectivity of ortho, meta, para isomers can also guideline for other separations. The enantiomer elution reversal with achiral derivatization is can also good tool for the enantiomer pure analysis for other enantiomeric pairs.

Are all the cited references relevant to the research?

Response: The cited papers are more or less relevant to the topic of this paper. We reduced the number of the cited literature in the corrected manuscript. In this way the more relevant papers are emphasized.

Reviewer 2 Report

The study of Repassy et al. "Structure – Chiral Selectivity Relationships of Various Mandelic Acid Derivatives on Octakis 2,3-di-O-acetyl-6-O-tert-butyldimethylsilyl-gamma-cyclodextrin Containing Gas Chromatographic Stationary Phase" aimed at the decision of a global problem of pharmaceutical science - chiral separation. For a long time this problem has attracting the attention of pharmaceutical scientists. But the challenges are still remained.

In the Introduction the authors thoroughly describe the background of the problem using the appropriate literature sources including several recently published articles. The purpose of the study is clearly stated in the end of the Introduction.

In Materials and Methods section the methods applied in the experiments and the equipment are carefully and clearly described.

The results of the study are well-presented and illustrated by the appropriate figures.

The intensity of inclusion phenomena was determined from the boiling points. Inclusion behaviors were shown to impact the chiral selectivity.

Only one chiral recognition mechanism was shown for the given separation.

The important conclusions are made from the obtained results. The impact of the nature and position of the substituents (structure) on chiral selectivity was proved.

The presented work is of a high quality and meets the scope of the IJMS. The authors solved all the problems stated in the purpose.

With minor corrections the manuscript may warrant publication.

1) Lines 259-261

Only one chiral recognition mechanism was estimated for all the studied enantiomeric pairs.

Can this fact be attributed to the structural similarity of the compounds? Please, give a short discussion on this issue.

Typos

Line 164

Remove round bracket.

Line 314

Add full point in the end of the sentence.

Line 324

"…the same…".

Author Response

Thanks for the carefull work of the reviewer.

Here are my responses in italics.

Suggestions for Authors

The study of Repassy et al. "Structure – Chiral Selectivity Relationships of Various Mandelic Acid Derivatives on Octakis 2,3-di-O-acetyl-6-O-tert-butyldimethylsilyl-gamma-cyclodextrin Containing Gas Chromatographic Stationary Phase" aimed at the decision of a global problem of pharmaceutical science - chiral separation. For a long time this problem has attracting the attention of pharmaceutical scientists. But the challenges are still remained.

In the Introduction the authors thoroughly describe the background of the problem using the appropriate literature sources including several recently published articles. The purpose of the study is clearly stated in the end of the Introduction.

In Materials and Methods section the methods applied in the experiments and the equipment are carefully and clearly described.

The results of the study are well-presented and illustrated by the appropriate figures.

The intensity of inclusion phenomena was determined from the boiling points. Inclusion behaviors were shown to impact the chiral selectivity.

Only one chiral recognition mechanism was shown for the given separation.

The important conclusions are made from the obtained results. The impact of the nature and position of the substituents (structure) on chiral selectivity was proved.

The presented work is of a high quality and meets the scope of the IJMS. The authors solved all the problems stated in the purpose.

With minor corrections the manuscript may warrant publication.

1) Lines 259-261

Only one chiral recognition mechanism was estimated for all the studied enantiomeric pairs.

Can this fact be attributed to the structural similarity of the compounds? Please, give a short discussion on this issue.

Response: The only one type of chiral recognitions mechanism refers for the given compound. Other test materials can have other chiral recognition mechanism but with only single one. The different tested enantiomeric pairs belong to different chiral recognition mechanism. The curved ln alfa – 1/T relationship show more than one chiral recognition mechanism simultaneously on a tested compound.

Typos

Line 164

Remove round bracket.

Response: corrected

Line 314

Add full point in the end of the sentence.

Response: corrected

Line 324

Response: corrected